# Molecular Regulation of the RhoGAP GRAF3 and Its Capacity to Limit Blood Pressure In Vivo

**DOI:** 10.3390/cells9041042

**Published:** 2020-04-22

**Authors:** Rachel A. Dee, Xue Bai, Christopher P. Mack, Joan M. Taylor

**Affiliations:** 1Department of Pathology and Laboratory Medicine, University of North Carolina at Chapel Hill, Chapel Hill, NC 27599, USA; rdee@email.unc.edu (R.A.D.); xue_bai@med.unc.edu (X.B.); cmack@med.unc.edu (C.P.M.); 2McAllister Heart Institute, University of North Carolina at Chapel Hill, Chapel Hill, NC 27599, USA

**Keywords:** GRAF3, RhoA, FAK, smooth muscle, blood pressure, hypertension, cardiovascular

## Abstract

Anti-hypertensive therapies are usually prescribed empirically and are often ineffective. Given the prevalence and deleterious outcomes of hypertension (HTN), improved strategies are needed. We reported that the Rho-GAP GRAF3 is selectively expressed in smooth muscle cells (SMC) and controls blood pressure (BP) by limiting the RhoA-dependent contractility of resistance arterioles. Importantly, genetic variants at the GRAF3 locus controls BP in patients. The goal of this study was to validate GRAF3 as a druggable candidate for future anti-HTN therapies. Importantly, using a novel mouse model, we found that modest induction of GRAF3 in SMC significantly decreased basal and vasoconstrictor-induced BP. Moreover, we found that GRAF3 protein toggles between inactive and active states by processes controlled by the mechano-sensing kinase, focal adhesion kinase (FAK). Using resonance energy transfer methods, we showed that agonist-induced FAK-dependent phosphorylation at ^Y376^GRAF3 reverses an auto-inhibitory interaction between the GAP and BAR-PH domains. Y376 is located in a linker between the PH and GAP domains and is invariant in GRAF3 homologues and a phosphomimetic ^E376^GRAF3 variant exhibited elevated GAP activity. Collectively, these data provide strong support for the future identification of allosteric activators of GRAF3 for targeted anti-hypertensive therapies.

## 1. Introduction

High blood pressure, known clinically as hypertension (HTN), is a highly prevalent and relevant disease in the Western world and is a major risk factor for myocardial infarction, stroke, and kidney failure. It is estimated that 1 in every 3 adults in the US has HTN and another 1 in 3 has prehypertension [1]. Despite the widespread availability of several classes of antihypertensives, including diuretics, ACE inhibitors, AngII receptor blockers, and calcium channel blockers, these drugs are usually prescribed empirically and are often ineffective [2,3]. In fact, it is estimated that only half of medicated adults have their blood pressure (BP) under control [4]. Thus, given the prevalence of HTN and its deleterious effects on cardiovascular outcomes, the identification of better HTN therapies could lead to a huge reduction in global cardiovascular disease burden.

While BP regulation involves the integrated control of many different organ systems and the interplay of many genes [5,6], one of the key components of HTN is increased peripheral vascular resistance due to the contraction of vascular smooth muscle within the arteriole wall [5,6,7,8]. Excitation-contraction coupling in smooth muscle cells (SMC) is mediated by the Ca^2+^-calmodulin dependent activation of myosin light chain (MLC) kinase and SMC tension is directly proportional to MLC phosphorylation [9,10]. SMC contractility is also enhanced by activation of the small GTPase RhoA which leads to direct phosphorylation of MLC by Rho-kinase (ROCK) and to ROCK-dependent inhibition of myosin phosphatase [11,12,13,14]. RhoA activity is also critical for de novo formation of actin filaments and formation of focal adhesions that are required for myosin-dependent force development and transmission, respectively. Finally, we and others have shown that RhoA activation promotes a robust positive transcriptional feedback circuit that activates a “contractile” SMC gene program. RhoA-dependent gene targets include integrin and RhoGTPase signaling molecules, cytoskeletal and contractile proteins, as well as many of the transcription factors that drive expression of these genes (e.g., SRF and the Myocardin factors) [15]. Thus, altered RhoA activity can have a major impact on SMC contractility and vessel tone [9,11,13] and of clinical importance, several lines of evidence have implicated RhoA signaling in the development of HTN. Increased ROCK activity has been observed in spontaneously hypertensive rats and some hypertensive patient populations [13,16], and ROCK inhibitors like Y-27632, Fasudil, and SAR407899 have been shown to reduce BP in hypertensive animal models and patients [17].

RhoA is activated by guanine nucleotide exchange factors (GEFs) that facilitate GDP-GTP exchange and is inhibited by GTPase Activating Proteins (GAPs) that facilitate RhoA’s intrinsic GTPase activity. Our recent collaborative efforts have demonstrated that the Rho-specific GTPase activating protein GRAF3 (GTPase Regulator Associated with FAK) is highly and selectively expressed in SMC in mice and humans and controls blood pressure by inhibiting RhoA-mediated SMC contractility. Specifically, we showed that GRAF3-deficient mice exhibit significant hypertension and increased pressor responses to Angiotensin II and endothelin 1; an effect that can be prevented by treatment with the Rho kinase inhibitor Y-27632 [18]. Both large and small arteries from GRAF3-deficient mice also exhibited increased contractility in vitro and in vivo [18]. We and others also reported that patient-specific polymorphisms in this gene are associated with human hypertension and we identified a causal mechanism by which a genetic variant controls this parameter [19,20,21,22]. As an exciting example of the clinical relevance of our work, Fjorder et al. recently identified a Danish family with early onset hypertension that had a chromosomal rearrangement in GRAF3 which led to haploinsufficiency [23].

In spite of the strong link between the RhoA signaling axis and the development of HTN, surprisingly few treatments are available that directly target this pathway. Herein, we developed and utilized a novel transgenic mouse line to show that modest ectopic expression of GRAF3 in SMC has the capacity to stably reduce BP. Importantly, we provide supportive evidence that endogenous GRAF3 protein toggles between inactive and active states that are controlled by post-translational modifications. Collectively these findings support the possibility that pharmacological targeting of GRAF3 could lead to a new class of therapeutics to treat hypertension and to reduce the morbidity and mortality associated with this disease.

## 2. Materials and Methods

### 2.1. Generation and Characterization of GRAF3^RQ^ SMMHC-CreER^T2^ Mice

Chimeric mice were produced in-house by injection of *GRAF3^RQ^* ES cells into the blastocoel cavity of mouse blastocysts by standard procedures. The *GRAF3^RQ^* strain was established using two independent chimeras that demonstrated germline transmission when bred to wild-type C57bl6 mice. *GRAF3^RQ^ SMMHC-CreER^T2^* mice were generated by crossing *GRAF3^RQ^* female mice with *SMMHC-CreER^T2^* male mice. All experiments were performed using age and sex-matched genetic controls. The *SMMHC-CreER^T2^* line is currently the most specific and robust SMC-specific Cre line available. However, because this BAC transgene was randomly incorporated into to the Y chromosome, we were limited to using male mice for our studies. Genotyping was performed using DNA isolated from tail biopsies using locus-specific primers (for *GRAF3^RQ^*: 5′-gttcggcttctggcgtgtgac; 3′-ggtccctcgaagaggttcactag, for *SMMHC-CreER^T2^*: 5′-tgaccccatctcttcactcc; 5′-aactccacgaccacctcatc; 3′-agtccctcacatcctcaggtt). Expression of the transgene was assessed by Western analysis and by quantifying levels of *GRAF3*, *GAPDH* (reference gene), *ACTB* (reference gene) and *SM22* (GRAF3 target gene) in bladders and aortas isolated from 8-month-old *GRAF3^RQ^ SMMHC-CreER^T2^* and genetic control mice. Forty-eight hours later after the last dose of tamoxifen, bladders were isolated and flash frozen while thoracic aorta segments were isolated and RNA was extracted using Qiagen RNeasy fibrous tissue kit (Germantown, MD, USA). Semi quantitative RT–PCR or quantitative RT-PCR as indicated was performed with the following primers: GRAF3 exons 1–4, 5′-CTGCCCACTCTGGAGTTCAGCG, 3′-GCTGCACCGATCTGTTCTTTTCG; GAPDH, 5′-ATGGGTGTGAACCACGAGAA, 3′-GGCATGGACTGTGGTCATGA; SM22, 5’-TGGGCGGCCTACATCAGGGC, 3’-CGGGGTGGTGAGCCAAGCAGA; ACTB, 5’-AGAGCTATGAGCTGCCTGACGGC, GGATGCCACAGGATTCCATACCC. Animal husbandry was provided by staff within the University of North Carolina Division of Comparative Medicine and all animal procedures were approved by our accredited American Association for Accreditation of Laboratory Animal Care committee and the Institutional Animal Care and Use Committee #329.

### 2.2. Blood Pressure Measurements

Conscious blood pressure was measured in male mice aged 12–16 weeks using radiotelemetry (Data Sciences International, New Brighton, MN, USA). Implantable mouse BP transmitters (PA-C10) were used to record arterial pressure in conscious and freely moving mice. In brief, the mice were anaesthetized with 2% isoflurane, the telemetry catheter was inserted into the left carotid artery of the mouse and the catheter tip was advanced into the thoracic aorta. The catheter was fixed in the left carotid artery and the transmitter was inserted subcutaneously along the right flank. Mice were allowed 7 days of recovery following transmitter implantation and were housed individually in a standard polypropylene cage placed on a radio receiver. Following baseline readings, mice were treated with tamoxifen (100 mg/kg) for 3 consecutive days via oral gavage. Increasing doses of Nω-Nitro-l-arginine methyl ester hydrochloride (L-NAME) salt (50 mg/L, 150 mg/L, 450 mg/L) (Sigma, St. Louis, MO, USA) were added to drinking water for 7 days (per dose). Mice were maintained in a 12:12 h light/dark cycle. All blood pressure parameters were telemetrically recorded and stored with the Ponemah data acquisition system (Data Sciences International, New Brighton, MN, USA). Recordings were collected for 5 min every 30 minutes throughout the study and averaged over a 24-h period for each day.

### 2.3. Cell Culture

Cos cells and rat aortic SMCs (RaAoSMCs) were maintained in DMEM (Gibco, Waltham, MA, USA) or DMEM-F12 media (Gibco, Waltham, MA, USA), respectively, supplemented with 10% fetal bovine serum and 0.5% penicillin/streptomycin. Cells were transfected with plasmids using Trans-IT (Mirus Bio, Madison, WI, USA) transfection reagents according to the manufacturer’s protocol. Myc-GRAF3 was made by cloning GRAF3 into a pCMV-Myc vector (Clonetech, Mountain View, CA, USA). Flag-GRAF3 Bar-PH was made by cloning into a pcDNA3 vector. GST-GRAF3-BAR-PH-GAP was made by in-fusion cloning (Clonetech, Mountain View, CA, USA) into a pGEX6.1 vector (GE, Marlborough, MA, USA). All phosphomimetic and phosphodeficient mutations were made by site-directed mutagenesis (Clonetech, Mountain View, CA, USA). Where indicated, cos cells were infected with LacZ/Cre adenovirus (Developmental Studies Hybridoma Bank, University of Iowa, Iowa City, IA, USA) for 24 h.

### 2.4. Molecular Modeling

Molecular models of GRAF3 were built using PyMol to combine the BAR-PH domains of Appl1 (PDB ID 2Q13, Human Appl1) and the GAP domain of GRAF1 (PDB ID 1F7C, chicken GRAF1). The domains from these proteins were chosen because they were the most similar and highly conserved proteins/domains (compared to GRAF3) that had solved experimental structures available on the Research Collaboratory for Structural Bioinformatics (RCSB) Protein Data Bank (PDB) (www.rcsb.org). The GAP domain was then docked onto the BAR-PH domain using the Clus Pro protein-protein docking server [24,25,26]. Models were narrowed down by analyzing the location and plausibility of residues important for GTPase binding and hydrolysis.

### 2.5. FRET Conformation Assay

Rat aortic SMCs were transfected with CFP-GRAF3-BAR-PH-GAP-YFP plasmid. The next day, cells were plated on a delta T dish (Bioptechs, Butler, PA, USA) and starved for 4 h prior to treatment with 10 μM Sphingosine 1 phosphate (S1P). CFP and FRET signals were excited at 440 nm and emission signals were collected at 470 nm and 535 nm, respectively, using an Olympus IX-81 inverted microscope. Image J software was used to calculate FRET signals. The FRET/CFP ratio of each cell was normalized to the FRET/CFP value before S1P stimulation.

### 2.6. Western Blotting

Cells or tissues were lysed in RIPA buffer + 0.5% Triton with protease and phosphatase inhibitors. Protein concentration was determined by using a colorimetric BCA assay (Pierce, Waltham, MA, USA). Lysates were electrophoresed on 10–15% SDS-polyacrylamide gels, transferred to PVDF membrane, and immunoblotted with specific antibodies as indicated using a 1:1000 dilution. Myc, Src, pMLC, MLC and GRAF3 pY376 antibodies were from Cell Signaling (Danvers, MA, USA); Flag antibody from Sigma (St. Louis, MO, USA); pTyr antibody from EMD Milipore (Burlington, MA, USA); GAPDH from Novus Biologicals (Littleton, CO, USA); GFP antibody from Meridian Life Sciences (Cincinnati, OH, USA). Blots were washed in TBST (TBS plus 0.05% Triton X-100), followed by incubation with horseradish peroxidase conjugated secondary antibodies (Amersham, Marlborough, MA, USA) at a 1:2000 dilution. Blots were visualized after incubation with chemiluminescence reagents (ECL, Amersham, Marlborough, MA, USA).

### 2.7. Immunoprecipitation

Cos cells were transfected with myc-GRAF3 variants and either Src or Flag-superFAK plasmid constructs. After 24 h, cells were lysed with radioimmunoprecipitation assay (RIPA) buffer + 0.5% Triton X-100 and myc-tagged GRAF3 was immunoprecipitated overnight from cell lysate using a myc antibody (Cell Signaling, Danvers, MA, USA) conjugated to Dynabeads Protein G (Invitrogen, Carlsbad, CA, USA). Immunoprecipitates and lysates were electrophoresed on a 10% SDS-PAGE gel and immunoblotted with indicated antibodies.

### 2.8. Protein Purification

GST-GRAF3 constructs were transformed into BL-21 competent *E. coli* and expressed by IPTG induction at 16 °C overnight. *E.coli* were lysed with a lysozyme-buffer (20 mM HEPES, 150 mM NaCl, 5 mL MgCl_2_, 1% Triton X-100, 1 mM DTT, 5 mg/mL lysozyme, protease inhibitors), sonicated and then protein was purified using glutathione-sepharose beads (GE Life Sciences, Marlborough, MA, USA). For GAP assay, constructs were dissociated from beads via reduced glutathione and GST was removed via PreScission Protease (GE LifeSciences, Marlborough, MA, USA) according to the manufacturer’s protocol.

### 2.9. Radioactive In Vitro Kinase assay

Purified GST-GRAF3-BAR-PH-GAP or GST-GRAF3 BAR-PH-GAP-Y376F were incubated in kinase buffer (25 mM MOPS, 25 mM MgCl_2_, 5 mM EGTA, 2 mM EDTA, 12.5 mM β-glycero-phosphate, 2.5 mM DTT) with either active FAK (R&D Systems, Minneapolis, MN, USA) or Src (Sigma, St. Louis, MO, USA) kinase and radioactive γ-^32^P (Perkin Elmer, Waltham, MA, USA) at 30°C for 15 min. Reaction mixture was run on a 10% SDS-PAGE gel, gel was dried and phosphorylation was assessed by radiograph. Gel was rehydrated and stained with Coomassie Blue to assess total protein concentration.

### 2.10. Time Resolved-Fluorescence Energy Transfer (TR-FRET) Assay

The Transcreener GDP TR-FRET Red enzyme assay was optimized according to the manufacturer’s instructions (BellBrook Labs, Madison, WI, USA). Ten μL of GRAF3 (0.2 nM) in a reaction buffer containing 50 mM Tris-HCl (pH 8), 150 mM NaCl, 5 mM MgCl_2_, 2.5 mM CaCl_2_, 1 mM DTT, 0.01% Tween-20 10 μM was dispensed into 96-well white solid bottom plates. Next, the reactions were started by the addition of 10 μL of 1X GDP Detection Mixture (10 μM GTP, 10 nM RhoA, 26.8 nM GDP HiLyte647 Tracer, 8 nM GDP Antibody-Tb, 1X Stop and Detect Buffer). After 30 min incubation on ice, the TR-FRET signals were measured at 670 ± 10 nm and 620 ± 10 nm on a FRET setting on a CLARIOstar (BMG Labtech, Cary, NC, USA) plate reader.

### 2.11. Bioluminescence Resonance Energy Transfer (BRET) Assay

A HaloTag^®^ (Promega) was added to the C-terminal of a pNLF1-N (CMV/Hygro) vector. Then WT or Y376E mutated GRAF3 BAR-PH-GAP was inserted between a NanoLuc^®^ (Promega, Madison, WI, USA) tag and a HaloTag^®^. RaAoSMCs were transfected with HaloTag^®^ control and either GRAF3 WT or GRAF3 Y376E tagged plasmids. BRET signal was detected 48 h later by the NanoBRET™ Nano-Glo^®^ Detection System (Promega, Madison, WI, USA). Dual-filtered luminescence signals were measured on a CLARIOstar (BMG Labtech, Cary, NC, USA) plate reader using a 450 ± 40 nm donor band pass filer and a 600 ± 10 nm long pass filter.

### 2.12. Statistics

Unless stated otherwise, all data represent at least three individual experiments and are presented as means ± standard error of the mean (SEM) or ± standard deviation (SD). Means of normally distributed data were compared by two-tailed Student’s t-test, one-way ANOVA (followed by Tukey’s post-hoc correction) or linear regression where indicated and statistical significance is reported as *p*-values. A *p*-value < 0.05 was considered significant. Sample sizes were chosen based on an extensive literature search and standard exclusion criterion of two standard deviations from the mean were applied. All statistics were calculated in Excel or GraphPad Prism8.

## 3. Results

### 3.1. Increased SMC GRAF3 Expression Reduced BP in Mice

Our previous data using GRAF3 hypomorphic mice clearly demonstrated that SMC GRAF3 is required for normal BP homeostasis [18]. Importantly, we have also shown that modestly increased expression of GRAF3 in cultured SMC reduced RhoA-dependent contractility as measured by decreases in MLC phosphorylation, actin polymerization, focal adhesion formation and SMC contractile gene expression [18]. Collectively, these data support the exiting possibility that increasing GRAF3 activity in SMC could be a promising strategy for inducing SMC relaxation and lowering BP.

To begin to test this possibility, we created a novel mouse model in which GRAF3 expression could be temporally and spatially induced in a Cre-dependent fashion using our well-characterized “stop and go” transgenic construct [27,28] (Figure 1A,B). To help ensure modest increases in SMC GRAF3 activity we utilized a GRAF3 variant (R417Q; denoted RQ). R417 is invariant in GRAF3 and studies in related RhoGAP family members have shown that an arginine in this position helps to align nucleophilic water to promote GTP hydrolysis [29,30]. Using an ELISA-based RhoA activity assay on immune complexes as previously described [18] we found that this variant has ~ 60% lower intrinsic GAP compared to Wt GRAF3 (4.2 ± 0.8 fold vs. 1.8 ± 0.1 fold for Wt versus RQ, respectively, vs. no GAP control, *p* < 0.05). As shown in Figure 1C, myc-GRAF3^RQ^ protein was induced 48 h following tamoxifen treatment of *GRAF3^RQ^ SM MHC-CreER^T2^* mice relative to similarly treated genetic controls. This resulted in an approximate 3-fold increase in *GRAF3^RQ^* over endogenous GRAF3 levels, as assessed by semi-quantitative RT-PCR using primers that recognize endogenous mouse and transgenic human *GRAF3* (Figure 1D,E). Importantly, in strong support of a functional role for the GRAF3^RQ^ transgene, expression of the RhoA-responsive SM contractile gene, *SM22* [31,32,33] was markedly reduced in the aortas from *GRAF3^RQ^* mice (Figure 1F). This finding confirms and extends our studies indicating that GRAF3 levels impart tight control over the expression of SMC differentiation genes that support contractility and suggested that transgenic expression of GRAF3 might confer a long-lasting reduction in vessel tone. To directly test this possibility, we implanted 12–16 week old male offspring with radio telemeters to continuously monitor BP in freely moving animals before and after GRAF3 induction. As expected, no significant difference was observed in basal blood pressure of un-treated *GRAF3^RQ^* and *GRAF3^RQ^ SM MHC-CreER^T2^* mice. However, 4 days following tamoxifen treatment, systolic BP in *GRAF3^RQ^ SM MHC-CreER^T2^* mice dropped significantly by ~10 mmHg (*p* < 0.05) relative to similarly treated *GRAF3^RQ^* mice (Figure 1G). Diastolic BP, mean arterial pressure and heart rate did not vary significantly between the two groups throughout the experiment (Appendix A). While GRAF3^RQ^ overexpressing mice were still amenable to BP increases when challenged with the hypertensive agonist L-NAME, it is of importance to note that these mice maintained a stable 10 mmHg decrease compared to their non-overexpressing counterparts.

Collectively, these studies indicate that SMC GRAF3 is both necessary and sufficient to lower BP and support the thesis that strategies to enhance GRAF3 could have therapeutic utility.

### 3.2. GRAF3 Activity is Dynamically Regulated by Auto-Inhibitory Interactions

While the aforementioned data provide important proof-of-concept that increased GRAF3 levels can reduce BP, the regulation of gene expression is not a reasonable approach for anti-HTN therapies. Likewise, small molecule regulators of protein-protein interactions (i.e., to target the GAP:RhoA interface) are not ideal for drug discovery. However, it is feasible to identify highly potent and selective compounds that mediate allosteric regulation of enzymes [34]. Interestingly, the enzymatic GAP activity of the GRAF3 family members, GRAF1 and oligophrenin are regulated by allosteric auto-inhibitory homo-dimeric interactions between the central GAP domains and the N-terminal BAR and PH domains [35]. Since GRAF3 shares the same major structural domains (Figure 2A), we postulated that it too might be controlled by allosteric inhibition. To begin to determine if GRAF3 is regulated in a similar fashion in SMC, we ectopically expressed full length GRAF3 with or without truncated variants containing only the BAR and PH domain. As shown in Figure 2B, ectopic expression of GRAF3 in primary vascular SMC markedly attenuated RhoA-dependent actin stress fiber formation but co-expression of GRAF3 BAR-PH mitigated this response—consistent with an auto-inhibitory mechanism (Figure 2C).

Since structural information for GRAF3 (or any other BAR-PH-GAP containing protein) is not yet available, we next employed molecular modeling strategies to gain insight into putative mechanisms that might control such auto-regulation. To this end, we first used PyMol to build a predictive molecular model of GRAF3 based on the solved crystal structures of the Appl1 BAR-PH domain [36] and the GAP domain from GRAF1 [30] using available data on the Research Collaboratory for Structural Bioinformatics (RCSB) Protein Data Bank (PDB) (www.rcsb.org) [37]. These domains were chosen because of their structural similarities to GRAF3 and their high conservation of functionally important or interface-interacting residues. We then performed a molecular docking search using ClusPro to dock two isolated GAP domains onto the BAR-PH domain dimer with a distance less than 30Å. Possible docking modes recovered 17 solutions that clustered into 2 groups of structures (Figure 2D,E). The structure on the top (Figure 2D) likely represents an open ‘active’ state because all of the residues important for GTPase binding and GAP activity (pink and dark blue, respectively) are exposed. The structure on the bottom (Figure 2E) represents a closed, ‘inactive’ conformation since the conserved face of the GAP domain is lying on the BAR-PH domain, effectively masking the residues required for GTP hydrolysis. Interestingly, the change in orientation of the GAP domain between the two dockings is a simple rotation about the horizontal axis by 90 degrees (note the different positions of the teal colored residues). Importantly, our analysis suggests that the GAP domain interacts with the convex surface of the banana-shaped BAR dimer. This finding is consistent with biochemical evidence that the BAR domain membrane binding occurs via interactions with lysines that lie on the opposite concave surface and that membrane binding does not interfere with GRAF1 or oligophrenin BAR-GAP interactions [35].

Using this structural information, we developed a novel GRAF3 biosensor to determine if GRAF3 undergoes dynamic allosteric regulation in response to physiologically relevant cellular stimuli. In brief, we engineered a truncated version of GRAF3 with a cyan fluorescent protein moiety (CFP) fused N-terminal to the BAR domain and a yellow fluorescent protein moiety (YFP) fused C-terminal to the GAP domain. Based on our model, the CFP and YFP moieties would be in close proximity (less than 20Å) when the protein is in a closed (auto-inhibited) state, thus permitting FRET between the two fluorophores; while relief of auto-inhibition would increase the distance between the fluorophores and prohibit FRET (Figure 3A). In strong support of our postulate that GRAF3.

Undergoes dynamic allosteric regulation in cells, we found that GRAF3 FRET/CFP ratios were spatially and temporally regulated in SMC following treatment with the contractile agonist, Sphingosine 1 phosphate (S1P) (Figure 3B,C). Spatially, it is notable that high levels of CFP (but lack of FRET) at the cell periphery 20 min following treatment, indicates that GRAF3 is most active in these protrusive areas. Temporally, total cellular levels of GRAF3-FRET activity (indicative of GRAF3 inhibition) transiently decreased following S1P treatment- a finding that could reflect binding to active RhoA which peaks within 2 min following treatment of SMC with S1P [33]. The sustained increase in GRAF3 FRET that was observed to occur 8–20 min following treatment is consistent with the possibility that besides activating RhoA through the RGS family of Rho-specific GEFs [38], S1P may also limit RhoA GTP hydrolysis by suppressing GRAF3 GAP activity. This postulate is consistent with our previously reported findings that depletion of GRAF3 leads to elevated and prolonged S1P-mediated activation of RhoA in SMC [18]. Notably, while RhoA activity returned to baseline 15 min following S1P treatment, GRAF3-deficient cells exhibited significant RhoA activity for up to 30 min [18].

### 3.3. FAK and Src-Mediated Phosphorylation of GRAF3 at Y376 Promotes Allosteric Activation

Since our FRET studies revealed that the temporal dynamics of GRAF3 allosteric modulation occurred in a time-frame consistent with post-translational modification, we next performed an in silico screen to identify putative phosphorylation sites in exposed regions that would be predicted to alter steric inhibition by the BAR-PH domain. We identified 15 putative phosphorylation sites that had a NetPhos 2.0 prediction scores of 0.9 or greater and that were positioned near the interface between the BAR-PH and GAP domains. One such site, Y376, is located in the 10 AA linker between the PH and GAP domain, is invariant in GRAF3 homologues and was previously identified as a phosphorylation site for pp60Src [39], which we confirmed (Figure 4A and Video S1).

Src and focal adhesion kinase (FAK) physically and functionally interact and because GRAF family members associate with FAK, we tested the possibility that GRAF3 was also a substrate for FAK. Indeed, we found that co-expression of GRAF3 with either constitutively active ^529F^Src or superFAK (a variant of FAK with increased catalytic activity [40]) led to robust tyrosine phosphorylation of GRAF3 as assessed by immunoprecipitation and Western blotting with a either a pan pTyr Ab or a specific pY376GRAF3 Ab (Figure 4B–E). Previous large-scale phosphoproteomic studies identified tyrosine residues 376 and 792 as potential Src-mediated phosphorylation sites [41]; however, our studies revealed that Y376 is the major site of phosphorylation. Indeed, as shown in Figure 4B–E, neither Src nor FAK induced significant phosphorylation of a ^Y376F^GRAF3 variant, while a ^Y792F^GRAF3 variant was phosphorylated to a similar extent as Wt GRAF3 (Appendix A). The capacity of both active Src and FAK to directly phosphorylate ^Y376^GRAF3 was confirmed using in vitro kinase assays with purified GST fusion proteins (Figure 4F,G).

Because the above findings indicate that GRAF3 is a FAK substrate and since once activated, FAK induces RhoA inhibition [42,43], we postulated that phosphorylation of Y376 might activate GRAF3 by relieving BAR-PH mediated GAP inhibition. To test this possibility, we first utilized a time-resolved fluorescence resonance energy transfer (TR-FRET) Rho GTPase assay that relies on direct immunodetection of GDP to assess the activity of a phosphomimetic ^Y376E^GRAF3 variant. As shown in Figure 5A–C, when combined with GTP-loaded RhoA, both purified Wt-BAR-PH-GAP and the Y376E variant induced dose-dependent decreases in TR-FRET, however the Y376E variant exhibited significantly greater specific activity. These findings are significant, because they are the first to show that phosphorylation of 376 is not only necessary but is also sufficient to promote GRAF3 activation. Moreover, these data suggest the possibility that FAK may down-regulate RhoA, at least in part, by activating GRAF3.

We next sought to test our postulate that Y376 phosphorylation activates GRAF3 by altering BAR-PH/GAP interactions. To this end, we modified our GRAF3 biosensor to include a bioluminescent donor that enables precise quantification of conformational rearrangements of proteins in cells as the donor, acceptor and bioluminescent resonance energy transfer (BRET) signals provide internal controls for protein expression [44] (Figure 5D). As shown in Figure 5E, a strong BRET signal was observed when WT GRAF3 was expressed in SMC while the maximal BRET signal induced by ^Y376E^GRAF3 expression was significantly lower. These findings are consistent with the idea that phosphorylation of Y376 induces activation of GRAF3 through an allosteric mechanism and highlight the exciting possibility that small molecules that stabilize the active conformation could prove useful as anti-hypertensive therapies.

## 4. Discussion

Vascular resistance is a major determinant of BP and is controlled, in large part, by RhoA-dependent smooth muscle cell (SMC) contraction within small peripheral arterioles [5,6,7,8,45]. Previous studies from our lab indicate that GRAF3 limits RhoA activity in vascular SMC and endogenous GRAF3 controls SMC tone (and dampens BP) by reducing SMC calcium sensitivity and restraining expression of the SMC-specific contractile proteins that support this function [18,19,46]. When combined with a growing body of evidence that patient-specific eQTLs in the *GRAF3* gene are associated with human hypertension [19,47,48,49], these studies suggest that GRAF3 might be an attractive target for the treatment of HTN. Our current findings that modest induction of GRAF3^RQ^ in SMC significantly decreased basal and vasoconstrictor-induced BP and that endogenous GRAF3 can be activated by phosphorylation-induced conformational changes provide strong support for the feasibility of GRAF3 as a druggable target.

Our new findings support the postulate that GRAF3 serves as a mechanical strain-sensitive rheostat to prevent excessive feed-forward activation of the RhoA signaling axis and to limit SMC contractility and BP. RhoA activity is elevated by graded increases in intraluminal perfusion pressure and RhoA-dependent SMC contraction is important to shield downstream capillaries from the damaging effects of high flow [50]. However, counter-regulatory means to control this so-called myogenic response are also important as exaggerated constriction can lead to tissue ischemia and high blood pressure. We previously reported that GRAF3 transcription is induced by the RhoA/MRTF/SRF signaling pathway that GRAF3 levels directly correlate with plasma volume (and intraluminal pressure) [19,46]. Consequently, depletion of endogenous GRAF3 exacerbated volume-overload induced increases in BP. Interestingly, increased intraluminal pressure also activates cell surface adhesion molecules including integrins and the integrin-associated kinase, FAK which we now show promotes a phosphorylation-dependent increase in GRAF3 GAP activity. Collectively, these studies strongly support a model wherein both GRAF3 levels and activity are induced in response to increased intraluminal pressure to counter-regulate the myogenic response.

The ability of induced SMC-specific expression of GRAF3 to lower BP in mice, adds to a growing body of literature supporting the promising clinical utility of inhibiting the Rho pathway for BP control [51,52,53,54]. Nevertheless, to date very few treatments are available to target this pathway. RhoA interacts with a variety of effector molecules that mediate SMC contractility (or other functions) including the Rho-associated coiled-coil domain containing protein kinases (ROCK I and II), the diaphanous-related formins (mDia1 and mDia2), protein kinase N, citron kinase, rhophilin, and the rhotekins I and II, among other enzymes [55]. Of these, ROCK has captured most of the attention in the hypertension field and indeed, ROCK inhibition has been shown to reduce BP and vascular resistance in many models [16,34,56]. However, there are concerns about the suitability of ROCK inhibitors as viable long-term treatments for systemic HTN because like many kinase inhibitors, treatment with these compounds leads to the rapid development of drug resistance. Moreover, the relative lack of specificity of most of the ROCK inhibitors coupled with the reality that ROCK I and ROCK II are ubiquitously expressed, makes potentially unknown side-effects a significant drawback [56,57,58,59]. Thus, it is formally possible that targeting GRAF3 might provide greater therapeutic benefit than targeting Rho kinase, as such an approach would block additional downstream pathways implicated in SMC contractility (i.e., mDia1 and mDia2, MRTFA etc.). Moreover, because GRAF3 is selectively expressed in SMC, drugs that regulate GRAF3 activity could result in far fewer side effects. In addition, the use of validated activators could provide ‘personalized’ approaches that would better target the underlying pathophysiology (i.e., to limit HTN in patients lacking the GRAF3 protective allele).

Targeting GAPs for therapeutic advances for cancer treatment has been largely overlooked because oncogenic mutations in Ras and Rho generally render GAPs ineffective in promoting GTP hydrolysis [60]. While this is not a concern with respect to development as anti-hypertensive therapies, challenges still exist because effective treatments would involve enhancement (not inhibition) of GAP enzymatic activity. Nonetheless, because RhoGAPs are multi-domain containing proteins that are regulated by a wide variety of post-translational modifications, several possibilities exist for allosteric GAP regulation. Indeed, the activities of several RhoGAPs including the GRAFs, OPHN1, β-chimerin, DLC1, and p50 Rho GAP are regulated by dynamic intramolecular interactions that facilitate auto-regulation [35,39,61,62,63]. Previous studies showed that the BAR and PH domains of ASAP1 and GRAF family members (GRAF1 and Oligophrenin) physically associate with the cognate GAP domain to sterically inhibit its function [35,64] and our studies herein reveal that GRAF3 activity is controlled by a similar mechanism. However, without further structural information, we cannot formally rule out the possibility that the BAR-PH domain is a competitive inhibitor of RhoA.

Interestingly, in our molecular models, the nucleophilic water coordinating residue, R417 lies on the RhoA interacting face of the GAP domain and is surface exposed in the open “active” model of GRAF3 and while it has reduced activity, is not expected to impact the active conformation. However, in our model of the inactive conformation, R417 is predicted to form an electrostatic pair with residue E245 on the BAR-PH domain. Interestingly, like R417, E245 is evolutionally invariant and is conserved in other GRAF family members, indicating the likely importance of this site. Thus, it is formally possible that the Q417 variant might disrupt the inactive conformation and that the ability to do so contributes in part to its capacity to promote RhoA GTPase activity in spite of its reduced capacity to coordinate the nucleophilic water molecule for GTP hydrolysis. It will be of future interest to determine the importance of such electrostatic pairs in the modulation of GRAF family member activities.

Moreover, we found that GRAF3 protein toggles between inactive and active states by processes controlled by the mechano-sensing kinase, focal adhesion kinase (FAK). Using resonance energy transfer methods, we showed that agonist-induced FAK-dependent phosphorylation at ^Y376^GRAF3 reverses an auto-inhibitory interaction between the GAP and BAR-PH domains. Y376 is located in a linker between the PH and GAP domain, is invariant in GRAF3 homologues and a phospho-mimetic ^E376^GRAF3 variant exhibited elevated GAP activity in vitro and in SMC. Collectively, these findings support the feasibility of GRAF3 as a druggable target.

While our data reveal that phosphorylation of Y376 promotes allosteric activation of GRAF3, there are likely additional post-translational modifications that contribute to its auto-regulation. Indeed, while our BRET assays indicate that the Y376 phosphomimetic variant significantly alters BAR-PH-GAP conformations, the BAR-PH deficient variant exhibits higher activity than the Y376E variant in our cellular assays (data not shown), indicating that phosphorylation of this site does not fully relieve steric inhibition. Moreover, these data do not rule out the possibility that phosphorylation of this site alters the affinity of the RhoA/GRAF3 interaction. Also, while we show that Y376 is the major site of FAK and Src-dependent GRAF3 phosphorylation, both our phospho-tyrosine immunoprecipitations and radioactive kinase assay data indicate that Src phosphorylates GRAF3 at additional sites. However, it is certainly feasible that additional cryptic sites were exposed in the purified truncated version of GRAF3 utilized for the in vitro kinase assays. Nonetheless, discovery-based mass-spectrometry data provided on PhoshoSitePlus database (phosphosite.org) predicts 4 potential tyrosine phosphorylation sites on GRAF3 including (Y142, Y376, Y792 and Y870). To date, tyrosines 376 and 792 have been identified as sites of post-translational modification in 244 and 642 independent mass spec studies, respectively, while Y142 and Y870 have each been reported once [41]. With respect to putative functionality, Tyrosine 792 is located in the proline/serine rich domain between the GAP and SH3 domains and thus is poised to effect SH3-mediated protein-protein interactions and intracellular localization [39]. However, it is formally possible that this phospho-site could promote GRAF3 activity since SH3 domain-mediated interactions have been shown to autoinhibit the membrane-binding capabilities of N-Bar and F-Bar domains in the related proteins endophillin and syndapin, respectively [65,66,67]. While our data reveals that this Y792 is not a major target for Src or FAK, it will be of future interest to determine if and how this or other post-translational modifications impact GRAF3 activity.

In summary, our findings presented herein and elsewhere provide strong support for a model in which GRAF3 serves as a mechanical strain-sensitive rheostat to tightly couple SMC tone to intraluminal pressure and that induced expression of GRAF3 has the capacity to have a long-lasting impact on BP control. As myogenic dysfunction has been linked to not only hypertension but also stroke, diabetes, and Alzheimer’s disease [50], it would be of future interest to explore the function of GRAF3 as a modifier of the pathological progression of such diseases. Moreover, our findings that GRAF3 activation can be induced by phosphorylation dependent de-repression of BAR-PH mediated steric interference, highlights the possibility that small molecules which specifically target the BAR-PH/GAP interface could prove useful as anti-hypertensive therapies.

## Figures and Tables

**Figure 1 cells-09-01042-f001:**
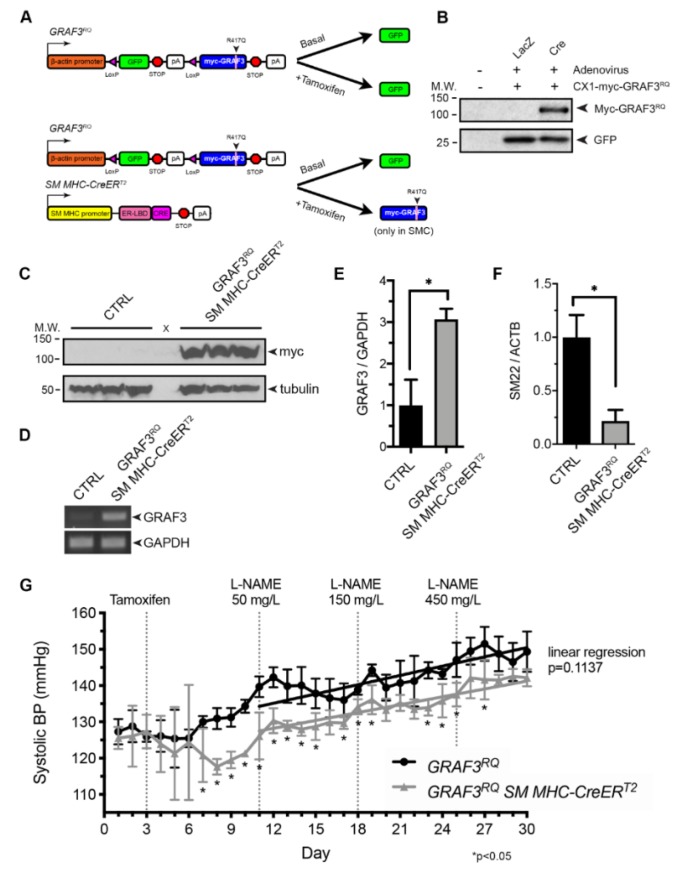
Smooth muscle cell (SMC)-specific GRAF3^RQ^ expression leads to a prolonged decrease in basal systolic blood pressure and limits hypertension (HTN). (**A**) Schematic of constructs used to develop tamoxifen-inducible SMC-specific GRAF3^RQ^ expression. (**B**) Western analysis of Cos cells transfected with the *GRAF3^RQ^* plasmid and infected with Cre or LacZ control virus. (**c**) Western analysis of bladder lysates from control and *GRAF3^RQ^ SM MHC-CreER^T2^* mice treated with tamoxifen (100 mg/kg for 3 consecutive days); *n* = 3 per group. (**D**) RT-PCR analysis of (**E**) *GRAF3* and *GAPDH* or (**F**) smooth muscle marker gene *SM22* and *ACTB* mRNA levels in thoracic aorta lysates from *GRAF3^RQ^ SM MHC-CreER^T2^* and genetic control mice treated with tamoxifen; *n* = 4 per group, **p* < 0.05. (**G**) Average 24-h systolic blood pressure, measured via radio-telemetry, of unrestrained, conscious *GRAF3^RQ^* and *GRAF3^RQ^ SM MHC-CreER^T2^* mice before and after tamoxifen treatment (100 mg/kg for 3 consecutive days) and increasing L-NAME doses (50 mg/L, 150 mg/L or 450 mg/L) given for a week (each) in drinking water. Data are expressed as mean ± SD; *n* = 4 for *GRAF3^RQ^* mice and *n* = 3 for *GRAF3^RQ^ SM MHC-CreER^T2^*; **p* < 0.05 vs. *GRAF3^RQ^* (Student’s t-test). Linear regression analysis was performed to compare the slope of the two groups after L-NAME treatment (*p* = 0.1137).

**Figure 2 cells-09-01042-f002:**
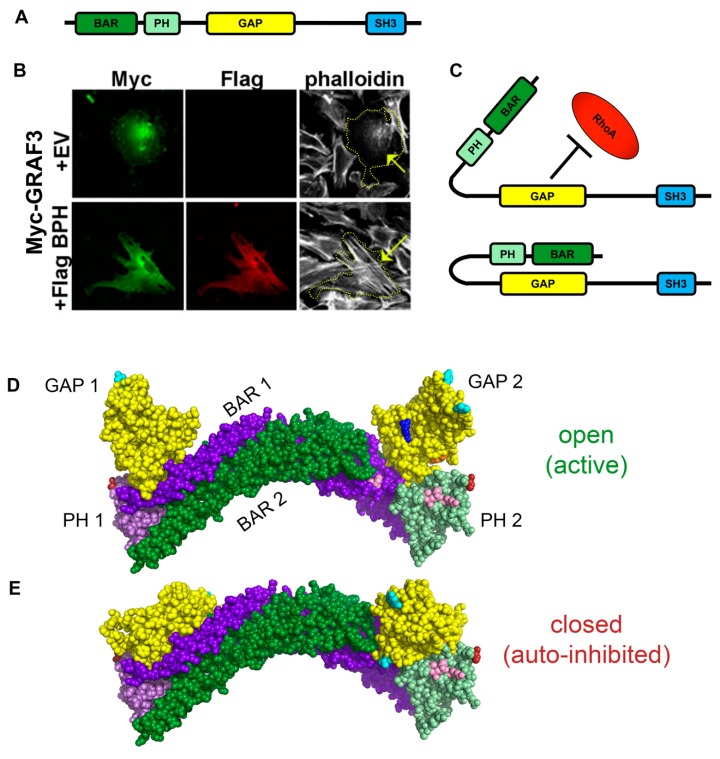
BAR-PH mediated autoinhibition of GRAF3. (**A**) Schematic of GRAF3 monomer domain structure. (**B**) Immunofluorescent staining of RaAoSMCs transfected with Myc-GRAF3 alone or Myc-GRAF3 co-expressed with Flag-BAR-PH (BPH) domain. Yellow arrow and dotted-outline indicate phalloidin-stained cell of interest that is positive for Myc- (green) and/or Flag- (red) staining. Data are representative of over 50 cells/condition from 3 separate experiments (58/59 GRAF3 expressing cells and 5/72 cells co-expressing GRAF3 and BAR-PH exhibited reduced stress fibers). (**C**) Schematic of GRAF3 in open (active) or auto-inhibited (inactive) conformations. Three-dimensional structures of the GRAF3 BAR-PH-GAP dimer were created using Pymol and the solved, similar structures of Appl1 (BAR-PH) and GRAF1 (GAP). ClusPro docking simulations predicted 2 conformations for GRAF3, (**D**) open and active or (**E**) closed and auto-inhibited. Color scheme follows: BAR 1 (dark purple), PH 1 (light purple), BAR 2 (dark green), PH 2 (light green), GAP 1 and 2 (yellow), arginine fingers (active site) (dark blue), RhoA docking sites (pink), C-terminus of PH domain (red), N-terminus of GAP domain (orange); residues in teal aids in visualizing rotation.

**Figure 3 cells-09-01042-f003:**
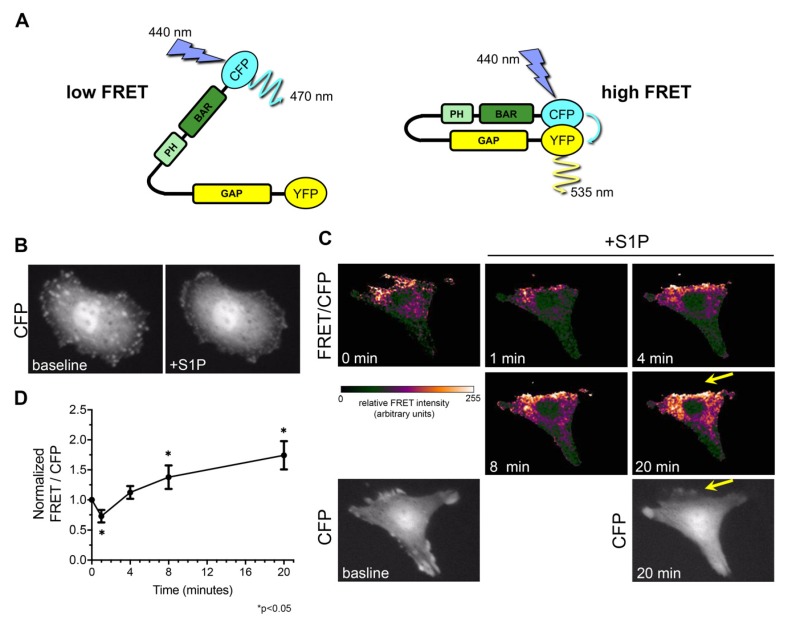
GRAF3 conformation is physiologically regulated in a spatial and temporal fashion. (**A**) Schematic of GRAF3 biosensor when there would be low and high fluorescence resonance energy transfer (FRET) signal. (**B**) RaAoSMC were transfected with the GRAF3 biosensor and CFP signal observed at baseline and 5 min after treatment with the contractile agonist S1P (10 µM). (**C**) FRET was monitored in RaAoSMC before and after treatment with 10 µM S1P. Note the dynamic temporal change in FRET and high levels of CFP (but lack of FRET) at the cell periphery 20 min following treatment, which indicates that GRAF3 is most active in these protrusive areas; yellow arrows. (**D**) Analysis of FRET/CFP ratio over time. Images are representative of 3 independent experiments with *n* = 7 cells per experiment. Data are expressed as mean ± SD; * *p* < 0.05 as assessed by one-way ANOVA and Tukey HSD. Each time point is significantly changed except for 0 min vs. 4 min (*p* = 0.11).

**Figure 4 cells-09-01042-f004:**
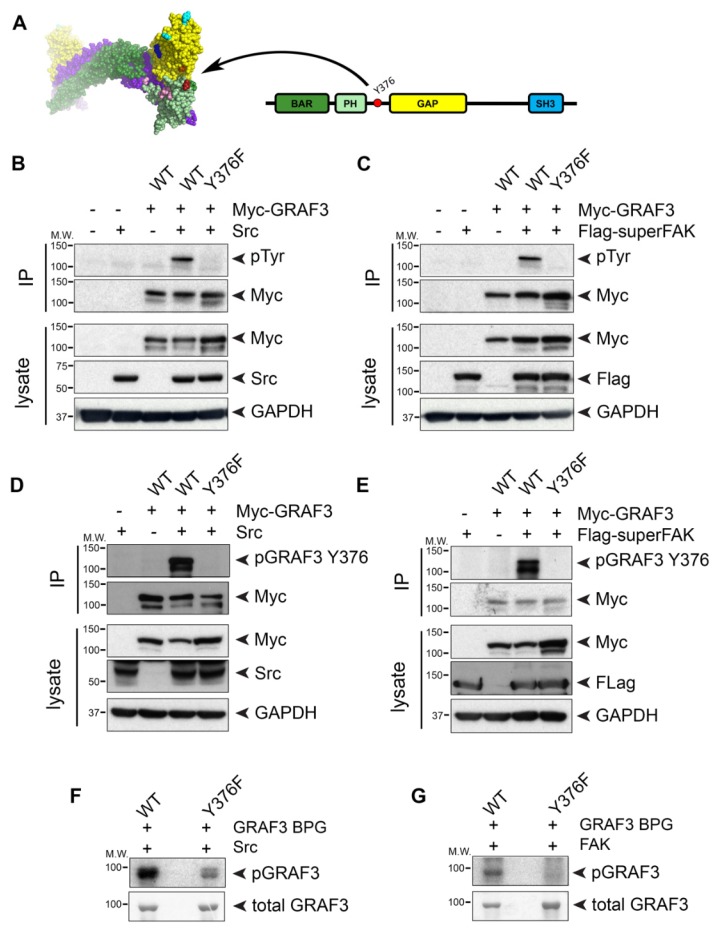
Both Src and FAK kinases phosphorylate GRAF3 at Y376. (**A**) 3-D (left) and 2-D (right) schematics of GRAF3 indicate location of Y376 in the unstructured, un-modeled linker between the C-terminus of the PH domain (red) and the N-terminus of the GAP domain (orange). (**B–E)** Cos cells were transfected with the indicated Myc-GRAF3 variant and either (**B**,**D**) ^529F^Src (Src) or (**C**,**E**) Flag-superFAK cDNAs. Myc-tagged GRAF3 was immunoprecipitated from cell lysate and blots were probed with indicated antibodies. (**F**,**G)** Purified GRAF3 BAR-PH-GAP (BPG) domain and ^Y376F^GRAF3 BAR-PH-GAP were subjected to a radioactive kinase assay using activated (**F**) Src or (**G**) FAK and ATP γ-^32^P. Phosphorylated and total GRAF3 are shown by radiograph (top) or Coomassie Blue staining (bottom), respectively. All blots are representative of 3 independent experiments.

**Figure 5 cells-09-01042-f005:**
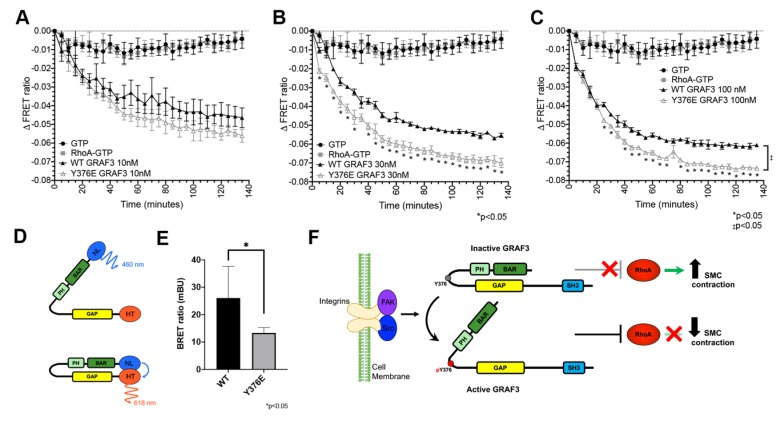
Phosphorylation of GRAF3 at Y376 increases GAP activity in vitro and decreases RhoA activity in SMC. TR-FRET GDP assay was performed using (**A**) 10 nM, (**B**) 30 nM and (**C**) 100 nM purified WT or Y376E GRAF3 BAR-PH-GAP variant. A decrease in signal indicates an increase in GTP to GDP conversion; n = 3, * *p* < 0.05 between WT GRAF3 and ^Y376E^GRAF3 at indicated timepoints, as assessed by unpaired t-test. Although not depicted, * *p* < 0.05 at all points except zero for both WT GRAF3 or ^Y376E^GRAF3 compared to either GTP or RhoA-GTP. GTP vs. RhoA GTP is not significant. One-phase decay linear regression revealed significance (‡ *p* < 0.05) between the slopes of WT GRAF3 and ^Y376E^GRAF3 at 100nM. (**D**) Schematic of GRAF3 BRET activation probe NL, NanoLuc. HT, HaloTag (**E**) GRAF3 WT vs. GRAF3 Y376E BRET activity in SMC. *n* = 6. mBU, miliBRET units. (**F)** Model of GRAF3 activation by Src or FAK phosphorylation of GRAF3 at Y376.

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
