# Peer review of "Molecular Regulation of the RhoGAP GRAF3 and Its Capacity to Limit Blood Pressure In Vivo"

_cells, 2020, doi:10.3390/cells9041042_

Round 1
Reviewer 1 Report
I feel that the authors have carefully considered my own - and those of the other reviewers - comments and have by re-phrasing and editing the manuscript and most importantly performing additional experiments to include more data further substantiated their findings. I would like to congratulate the authors on their study and fully support publication of the manuscript in its current form in "Cells".
Reviewer 2 Report
The authors have done well in addressing all my issues regarding the quantification of data and this reviewer thanks them for their effort. I feel the manuscript is greatly improved for removing the preliminary small molecule screen, it now has a more logical title that reflects better the main content of the study.
Reviewer 3 Report
No comments.
The authors have answered all my previous questions and criticism
This manuscript is a resubmission of an earlier submission. The following is a list of the peer review reports and author responses from that submission.
Round 1
Reviewer 1 Report
The submitted manuscript by Dee et al. is undoubtedly a comprehensive research study examining the role of GRAF3 in RhoA-dependent SMC contraction and regulation of blood pressure and validating GRAF3 as a potential drug target. It evidently is a well planned and executed studies, and the authors combined numerous sophisticated state-of-the-art techniques to address their research question. In general, their conclusions seem well supported by the data presented.
While I think this conceptually sound study is well suited for the special issue in „Cells“, the authors should address a few matters:
- For this study, the authors generated a novel mouse model that allows SMC-specific ectopic expression of a GRAF3 variant. The strategy to use a mutant with intrinsically lowered GAP activity is undoubtedly a smart move to limit overexpression effects and, phenotypically, the model presents as expected. Mentioning the R417Q mutant, the authors refer to activity assays or the like in ref. 18 (Bai et al. 2013), yet this variant is nowhere to be found in that manuscript. Given that the manuscript further dives into the finer details of GRAF3 GAP regulation and uncovers its allosteric regulation through phosphorylation, it would be interesting to hear more about the nature of the mutant. How does the mutant heave in the conformation (Fig. 3) and GDP (Fig. 5) assays?
- In line with this point, it would be interesting to incorporate the mouse model again towards the end of the study. For instance, is there evidence of GRAF3 phosphorylation in vivo? Is it altered in the mouse model?
- Figure 1, Panel B: In this panel, the myc-GRAF3 construct used to generate the SMC-specific transgenic mouse line is tested – but in transiently transfected and then Cre-treated cells. While this demonstrates the construct is functional, this figure still lacks the critical control experiment that demonstrates myc-GRAF3-RQ expression in those animals displaying the BP phenotype depicted in panel C. Western blot analysis of GFP and myc-GRAF3 in primary SMCs is required, a staining of a tissue section would be even more convincing
- Figure 2, Panel C: Changes in Phalloidin staining are not that easy to acknowledge; an overlay image of Myc and phalloidin would help the reader to directly notice that changes are to be observed in transfected cells; quantitative data given in the figure legend could also be put into a graph next to panel B to further strengthen the author’s observation
- Figure 4: As the authors discuss the relevance of GRAF3 Tyr phosphorylation in addition to Y376, it would be good to repeat experiments depicted in panel F and G using additional Tyr mutants and/or double mutants; evidently the Y376F mutation does not completely abrogate GRAF3 phosphorylation – or can the discrepancy between cell culture experiments (B-E) and F/G be explained
- inconsistent use of double-spacing should be corrected
Reviewer 2 Report
In this study by Dee et al the authors report on the functional role of GRAF3 as negative regulator of GTPase RhoA and its importance in the regulation of hypertension. The study aims to further characterise the molecular mechanisms required for activation of GRAF3, with a view to validate it as a druggable target to develop novel therapies for hypertension.
The study is well written, references correctly the scientific literature and shows some novel findings using biosensors and biochemical experiments to investigate the molecular regulation of GRAF3 via phosphorylation and its role in auto-inhibition of its GAP domain. The study ends with a small molecule screen for compounds that stimulated Rho-GTP hydrolysis in GRAF3 dependent manner.
The paper reads nicely, data presentation is ok, however I have few concerns over quantification of some of the experiments that needs improvement. This would strengthen the paper and give a more convincing data to support the authors conclusions.
Another main point is the small molecule screen at the end, is this required in this study? While I appreciate the effort gone into generate this data, if the authors are unable to reveal anything about what these small molecules are then it is somewhat detracts from the study and looks like the beginning of another paper.
Major points
1
In Figure 2 please could you add some quantification to this experiment showing % of cells that rescue the loss of stress fibres is this statistically significant?
2
In Figure 3 this data really needs quantitation to be more convincing. This could be done by measuring the amount of FRET at a region of interest say the cell edge at time 0 vs different times after stimulation with S1P. You probably have the numbers (7 cells from 3 independent experiments.)
3
In figure 5 would it not be and more consistent to look at F-actin staining as a readout of RhoA activity as in Figure 2 as well as morphology. The graph in G is bit confusing is there a significant difference in what category?
Minor points
A schematic drawing of the of FRET biosensor would be helpful to put alongside the structure in Fig 3 A
Line 351 “FAK activity dampens RhoA activation” what do you mean by this suggest rewriting to make clearer.
Reviewer 3 Report
The goal of the Dee et al. paper was to validate GRAF3 as a druggable target for high blood pressure. They developed various tools and technics to reach their this objective, and did a significant amont of work. However, both major and specific comments are listed below:
Main comments:
The main issue is that the paper lacks of focus. There is a succession of tool and technical developments, approaches and corresponding results (mice, in silico modeling, biosensor, TR-FRET, BRET probe, …) placed end to end but what is the message? What exactly is the subject of the paper? What do the authors want to address? What novelty do they want to point out?
The results are rather over-interpreted and the novelty and originality of the document is based more on the tools developed than on the results. Consistent with that, the title does not fit with the paper content.
1) The mouse line generated GRAF3RQ SM MHC-CreERT2 mice +/- tamoxifen is insufficiently described.
The causal relationship between GRAF3 activity and the BP phenotype is not demonstrated: expression of myc-R417Q-GRAF3 in vascular smooth muscle cells / arterial wall after tamoxifen treatment should be demonstrated (GRAF3RQ /MHC-CreERT2 mice vs GRAF3RQ mice), the "overexpression" of GRAF3 (endogenous GRAF3 + myc-R417Q-GRAF3) and the activity of RhoA in the arterial wall should be shown.
Finally, these data seem to be quite disconnected from the rest of the document.
2) Figure 2B shows that the ectopic expression of GRAF3 in SMC decreased actin stress fibres, suggesting that it down-regulated RhoA activity and thus its GAP activity led to GTP hydrolysis. Expressed GRAF3 is therefore not in an auto-inhibited state.
The observed inhibiton of this effect when GRAF3 was co-expressed with GRAF3 BAR-PH is not obligatory consistent with an auto-inhibitory mechanism. It can be interpreted as a competitive inhibition of RhoA binding to GRAF3 protein. Fig. 2C is not supported by the results shown in Fig. 2B.
3) According to the construct description, the closed (auto-inhibited, inactive) state permit FRET: A decrease in FREt signal is thus supposed to indicate an opening of the protein attesting its activation.
Fig. 3C shows a transient decrease (max at 1 min) of FRET signal after stimulation of RaAoSMC by S1P, indicating GRAF activation. It would be clearer to comment that "As expected, GRAF activity was correlated to RhoA activity, ….". However, this is not trivial. Why was it expected ?
4) Y376 has been already identified as an activating phosphorylation site that is phosphorylated by Src. Results shown here as mainly confirmatory, expect for the FAK-mediated Y376-GRAF5 phosphorylation.
5) Figure 5A-C. It is surprising that the slope of the curve for the 20 first minutes is the same for the WT and the Y376E-GRAF3. GAP activity are generally assessed for period ranging from 30-60 min, sometime less. What is the significance of this late difference? What is the relevance of this kinetics of GTP hydrolysis on RhoA when compared to physiological regulation of RHoA activity?
It is concluded a grater activity of Y376E-GRAF3 but an increased affinity of the RhoA/GRAF3 interaction would led to the similar observation.
One limitation of these experiments is the unique use of a mutant (Y376E-GRAF3), wich is believed to be phosphomimetic. The direct demonstration that Y376 phosphorylation stimulated GRAF3 activity has been already published (ref 33). The novelty only relies on the tools and the technical approach used.
6) No statistics in Fig. 5G?
It is written: "Consistent with our findings that purified Y376E-386 BARPHGAP induced higher GAP activity, a higher percentage of SMC expressing full length Y376EGRAF3 exhibited an arborized phenotype when compared to those expressing Wt GRAF3 (Figure 388 5G)": However, the graph Fig. 5D does not show any difference between WT and Y376E bars.
7) Results on the small molecule regulator screening is too preliminary.